# The initial development and validation of the Social Adaptability Skills Questionnaire: SASQ

**Samuel Owiti** *, **Denis Hauw**

Institute of Sport Studies of the University of Lausanne, Lausanne, CH Switzerland

* samuel.owiti@unil.ch

## Abstract

Changing clubs over the course of an athletic career may not always be easy, and this has raised questions about how these changes affect career development. However, few studies have focused on the process of adapting to a new club and the factors that lead to success or failure. To address this gap in the literature, we aimed to develop and provide the initial validation of a questionnaire designed to assess athletes' social adaptability skills (SAS). To do so, we conducted four studies, from the initial development stage to the final validation stage. In the first phase, we generated questionnaire items with clear content and face validity. The second phase explored the factor structure and reliability of the Social Adaptability Skills Questionnaire (SASQ). This was carried out with 543 young athletes in talent development through exploratory factor analysis (EFA), which was validated with confirmatory factor analysis (CFA). The EFA yielded a 17-item, four-factor structure with good internal reliability ( = 0.876). The CFA revealed that the model fit indices were acceptable (RMSEA = 0.06, CFI = 0.809, TLI = 0.844, and GFI = 0.926). In addition, Latent Class Analysis (LCA) was applied to determine the predictive validity of SASQ resulting into identification of three classes (low achievers, average achievers, and high achievers) with four discriminating dimensions (coach, teammates, family, and club). The SASQ appears to be a promising psychometric instrument of potential usefulness for education and program reviews in applied settings and a measurement tool in talent development research.

## Introduction

It is unusual today for athletes in team sports to spend their entire careers in the same club. In 2019, the CIES Observatory [1] reported that the number of international transfers in basketball alone rose 34% between the 2010/2011 and 2018/2019 seasons. The effects of these club-to-club moves on players' equilibrium and their sporting lives are not well known, although recurring individual stories (e.g., Mercato transfers in soccer, NBA basketball drafts, and major hockey league transfers) and a recent study have underlined distinct periods of adaptation to the new environment [2]. Although professional athletes may show adequate resources for adaptation as they move from club to club over the course of their careers, this may not be the case for talented young athletes preparing to enter the professional world, as they may lack some of the basic skills and psychosocial resources (example: positive thinking, goal setting skills, self-

**Data Availability Statement:** All relevant data are within the paper and Supporting Information files

**Funding:** The author(s) received no specific funding for this work.

**Competing interests:** The authors have declared that no competing interests exist.

confidence skills) that are needed to ensure successful moves [2]. However, in talent development research programs [e.g., 3–5] and especially those focused on team sports – where club changes are usual [2] – this issue has not been taken into account. We thus sought to develop a valid instrument to assess the psychosocial skills of young players that will be useful for future positive moves. We assumed that this instrument could also be used to design individualized programs to develop these skills in training centres for professional team sports.

The notion that young talented players also need to be equipped with psychosocial skills before they enter the professional world has thus been identified as a major concern. For example, Hauw and colleagues e [6] used the MacAdams identity multilayer model and identified three types of resources available to players during their transitions: (i) level 1: dispositional traits, which are general dispositions that can facilitate adaptations during club-to-club changes; (ii) level 2: personal action constructs (PACs), which draw on skills like personal strivings; and (iii) level 3: narratives, which are the life stories constituted from experience to give meaning to oneself. Regarding club changes, level 1 refers to the more or less high level of the personality disposition to new social environments (openness to experience, agreeableness, emotional stability, extraversion, and conscientiousness) [6]. The development of this level may require deep psychological work with young team athletes. Level 2 of the MacAdams model thus may be the right target for preparing players for future club changes. Level 3 refers to the life stories around the experience of club changes and thus might be used by sports psychologists when working with new professional athletes arriving in a club after experiencing other changes in their careers.

In the development of talented young athletes, studies have identified general competences such as self-regulation, goal setting, performance assessment, resilience, grit, and self-awareness as important for success [7–9]. MacNamara and colleagues drew up a list of ten key skills, which they referred to as the psychological characteristics of developing excellence (PCDEs), with positive or adaptive characteristics such as goal setting, self-organization, commitment, imagery use, self-awareness and others [10]. However, all these psychosocial skills are "situated resources", meaning they are attuned to specific and typical situations like training, competing, or relationships with coach or parents [11]. The need to identify the specific skills for successful club changes thus emerged.

Acculturation and integration studies indicate that most youths in host countries and cities face challenges during transitions, and this has prompted calls to facilitate their adaptation and ensure their full inclusion [12]. For example, these youths may need to learn the customs and language of the receiving country so that they can communicate and socialize to achieve full social inclusion [13]. Social skills are not only key factors in achieving inclusion but also a concern on the sports field, especially with young players. Therefore, instruments like the Social Skills Questionnaire (CHASO III) [13] and the Social Skills Questionnaire for College Students (SSQ-U) [14] have been used to assess these skills. However, these questionnaires are general or specific to college and not well situated for sports, especially club changes. For athletes, some of the elements of the situation that provide them affordances in the sporting environment are coaches and teammates' specific behaviours, as reported in several studies [2, 15, 16]. We therefore saw a need for a specific instrument that is situated, short, and quickly and inexpensively applied to measure the skills deployed during club change.

Research to date has reported that athletes' psychosocial skills influence their development in a variety of ways, including supporting (e.g., athletes' stability and harmony), constraining (e.g., undermining the well-being of players), accelerating (i.e., increasing performance success and satisfaction), or modifying how they handle problematic situations [2]. As part of their investigation into this problem, Owiti and Hauw [2] identified a range of psychosocial factors, known as social adaptability skills (SAS), that contributed to successful outcomes. These SAS

structures were organized into four factors (i.e., coach, teammates, family/friends, and club). The sub-themes in relation to the coach were: *obeying orders*, *language barrier*, *inequality/ favouritism*, *reduced play time*, *difficulty combining sports and other life domains*, *pressure to perform*, *lack of structured training*, and *low-level competition*. Sub-themes related to teammates included: *respect*, *language barrier*, and *negative perception of others*. This was followed by sub-themes related to family/friends, which included: *climate change*, *distance*, *language* and *culture shock barrier*. Last, club included the following sub-themes: *poor facilities and logistics* and *lack of support*. These results indicate the events of critical situations that can be encountered during club changes. Aspiring athletes must be prepared to deal with these situations in order to maximize their development opportunities (e.g., adapting during the club change), endure setbacks (e.g., adapting to coaching styles), and navigating key transitions (e.g., living away from family and friends) [2].

It is generally expected that young athletes possess the required psychosocial skills for sports participation and the pursuit of excellence. Yet there is a need for talent development programs to also consider the skills necessary for changing clubs and adapting to new environments. Having identified the key skills needed to adapt to different situations during club to club change, the next step is to develop a psychometrically sound measure of these skills with an ultimate goal of designing and testing a learning program for these athletes. The aim of the current research was thus to develop and validate a questionnaire to assess the level of SAS possessed by young athletes. Such an instrument is needed to provide regular feedback and refinement and to identify areas that require immediate attention, improvement, or maintenance. As a result of these findings, training programs could be formulated to improve athletes' SAS performance by identifying the challenges they face and including specific training to help enhance those weaknesses. Finally, the validation of this questionnaire would contribute significantly to the international research community focussed with talent development [2, 15].

To summarize, our aim was to outline the initial development and validation of a questionnaire designed to assess the SAS of young athletes. We thus designed four studies from the initial development stage up to the final validation stage.

## Study 1

The purpose of this first study was to generate the questionnaire items for the development of a psychometric measure of SAS that would be useful to athletes during club changes. Similarly to previous studies, the initial items were developed in accordance with the recommendations for developing new measurement scales [e.g., 17, 18].

### Method

The items were developed in several stages and elite basketball players were recruited for the first stage since this population changes clubs frequently over the course of their career progression [1].

**Participants.**   After receiving approval from the Research Ethics Commission of the University of Lausanne (Project number: E_SSP_122021_00002) the participants gave their written informed consent. The study involved a convenience sample of 20 European and American professional basketball players (age range: 20–36 years; mean: 26.05, SD: 4.12). Seventeen were male and three were female. In total, 12 players had attended European basketball academies in their respective countries and had performed at the highest national levels (e.g., Swiss Basketball League and National Basketball League of France). The eight others had gone through North American academies (e.g., in the USA & Canada) playing in one of the highest

leagues (e.g., the National Basketball Association G League and Women's National Basketball Association).

**Data collection.**   Individual interviews were conducted using a semi-structured interview guide. We sought to help the players recall their personal experiences during club changes. As part of the interview, we worked first on creating a timeline of the changes, which required us to break down the flow of time into periods to keep a semiotic trace of the adaptability experience.

The interview guide incorporated all of the changes from club to club to facilitate our in-depth understanding of the dynamics of personal experience within the context of each club's environment. All interviews were recorded digitally in their entirety and lasted from 30 to 90 minutes.

**Data analysis.**   Of the 20 participant interviews, we chose only episodes that displayed the problematic situations faced by the players during club changes. Identifying a problematic situation involved looking for disturbances in the transition that prevented players from performing at their best [2]. Data were analysed qualitatively using inductive content analysis. A constant comparative method was employed, which involved the development of additional codes, identification of emerging themes within the data, and continuous comparison of the codes (e.g.,, between participants or between time periods). In accordance with the criterion of data saturation [19–22], data collection and analysis were discontinued when the categories upon which the theory was built no longer producing new insights.

**Results.**   Four major themes consistently emerged from this analysis and comprised problematic meaningful experiences with (i) the coach, (ii) the teammates, (iii) being away from family and friends, and (iv) the club.

We therefore based our initial item generation on the four emerging themes and this resulted in a preliminary list of 55 items representing the 15 SAS sub-themes earlier mentioned.

**Item justification.**   *Expert panel.* An independent panel of six experts (*PhD and postdoc students in sports psychology)* reviewed the initial list of 55 items [4, 23, 24]. In addition to their expertise in psychological research, applied talent development, and coaching, all of the experts were familiar with the SASQ's aims and rationale. Each expert rated the content relevance and representativeness of each item on a scale from 1: *not at all relevant* to 5: *completely relevant*. Items that were rated 4: *relevant* or less were discussed by the whole panel [4]. At the end of this stage, some of these items were marked for deletion. A first panel review resulted in the rewording of several items ($N = 41$) due to grammatical errors and comprehension issues. A total of ten items were deleted and five additional items were added. Furthermore, the five items that were added were part of the interviews and not from the expert panel. A second expert panel was constituted and asked to follow the same procedure, and this resulted in the rewording of several items ($N = 25$) and deletion of several others ($N = 14$). As a result of this stage, 36 items were included in the SASQ.

**Cognitive interviews.**   During the third stage, cognitive interviews were conducted to check for misunderstandings, unclear questions, inconsistencies, and inappropriate options [4, 25]. A written informed consent was obtained from all participants (and parents/guardians when participants were under 16 years of age). Six young athletes in development were interviewed individually, reflecting the intended target population of the questionnaire (there were two participants in each age group: 12–14 years, 15–17 years, and 18–21 years). Two of the participants competed at a national level, whereas the remaining four competed at a regional level in their respective sport (football $N = 2$; basketball $N = 2$; handball $N = 1$, and ice hockey $N = 1$).

In each item, comments were directly entered under the question to summarize the findings. A complete review of the comments was conducted using Conrad and Blair's [25]

classification system. This process resulted in 34 items, of which two items were omitted due to lexical problems [26].

**Pilot test.** The fourth stage was a pilot test conducted using the 34-item version of the SASQ to examine the comprehensibility of the questionnaire and the ease of overall administration [4, 27, 28].

**Participants.** The 34-item list was pilot-tested with a sample of athletes ($N$ = 21, 12 males, 9 females; age range: 12–21 years) competing in different sports (basketball, $N$ = 9), (football, $N$ = 6) (rugby, $N$ = 3), and (handball, $N$ = 3). Each athlete competed at a representative level (e.g., regional and national levels) in their respective sports. In order to ensure that the SASQ was suited to the range of sports and age groups, a purposefully stratified sample was selected.

**Procedure.** After receiving approval from the Research Ethics Commission of the University of Lausanne (Project number: E_SSP_122021_00002) and informed written consent from all participants (and parents/guardians when participants were under 16 years of age). The pilot version of the SASQ contained 34 items using a 6-point Likert scale with a similarity response format from 1: *very unlike me* to 6: *very like me* as suggested by Lei Chang [4, 29]. The 34-item list contained a mixture of negatively ($N$ = 13) and positively ($N$ = 21) worded questions to minimize the danger of acquiescence bias [4]. The pilot version took between 12 and 15 minutes to complete. After the item analysis, participants were encouraged to write comments next to any problem items. Debriefing was conducted after each item on the questionnaire was completed and participants were encouraged to provide feedback.

**Data analysis and results.** Following the pilot test data analysis, the SASQ still contained 34 items, ensuring that at least two items per sub-theme were retained [30]. As a result of the first stage, the elite athletes played different roles, which contributed to the retention of all items, since they suggested themes focusing on their challenges during club to club changes. Young athletes were able to identify incomprehensible or difficult questions through the pilot test study. A revised questionnaire structure was sent to two members of the expert panel with the request that they comment on the face validity, content validity, comprehensibility, and comprehensiveness of the SASQ in light of the proposed target population [4]. This was done to ensure that the 34 items adequately covered the questionnaire constructs. The questionnaire did not require any further modifications after this stage.

## Study 2

In study 2, we performed exploratory factor analysis (EFA) to examine the possibility of reducing the large number of SASQ variables into manageable size while retaining as much of the original information as possible [31, 32].

### Method

**Sample size.** The Kaiser-Meyer-Olkin (KMO) test of sampling adequacy and the Bartlett sphericity test were used to assess the quality of the correlations in order to proceed (or not) with the EFA [33, 34].

**Participants.** A total of 543 participants (mean age: 16.67, SD: 2.42, range: 12–21 years) completed the questionnaire (see Table 1). There is no consensus regarding sample size calculation using either EFA or CFA [35]. Following considerable discussion within the literature, the current study followed the recommendation of DeVellis [35] indicating participant item ratio of 10:1. Therefore, since SASQ contained 34 items, we required between 340–510 to fulfil the empirical rule.

Participants were purposefully sampled in line with their characteristics and ability to adapt in the environmental sporting conditions for which the SASQ was designed. By virtue of their

**Table 1. Demographic characteristics of participants.**

| Participants characteristics | Total number | Frequency (%) |
|---|---|---|
| **Gender** | | |
| Male | 365 | 67.2 |
| Female | 178 | 32.8 |
| **Age** | | |
| 12yrs | 33 | 6.1 |
| 13yrs | 35 | 6.4 |
| 14yrs | 44 | 8.1 |
| 15yrs | 66 | 12.2 |
| 16yrs | 51 | 9.4 |
| 17yrs | 89 | 16.4 |
| 18yrs | 104 | 19.2 |
| 19yrs | 52 | 9.6 |
| 20yrs | 41 | 7.6 |
| 21yrs | 28 | 5.2 |
| **Sport type** | | |
| Basketball | 136 | 25 |
| Football | 173 | 31.9 |
| Volleyball | 70 | 12.9 |
| Unihockey | 39 | 7.2 |
| Handball | 66 | 12.2 |
| Rugby | 59 | 10.9 |
| **Talent development model** | | |
| Academy | 252 | 46.4 |
| Not in Academy | 291 | 53.6 |
| **Level of play** | | |
| National | 126 | 23.2 |
| Regional | 417 | 76.8 |

commitment and selection to formally established development environments (e.g., regional or national, and in academy or not in academy groups), they had been recruited as junior athletes with the potential to become senior elites.

**Procedure.** Approval was received from the Research Ethics Commission of the University of Lausanne (Project number: E_SSP_122021_00002) and written informed consent was obtained from all participants. Self-administered paper-pencil SASQ questionnaires were distributed to the 543 participants at their respective clubs between March and May 2022. As participation was anonymous, the athletes were given coded names based on their initials and age and their parents' initials (mother then father).

**Data analysis.** The Statistical Package for Social Sciences (SPSS *version 27*) was used to examine the factor structure of the SASQ with statistical significance set at $p < 0.05$. Prior to the analysis, scores of negatively worded questions were reversed. Descriptive statistics were performed on the 34 items with means, standard deviation, measures of kurtosis and skewness, and the analysis of missing values. Participants could decide not to answer certain questions. For the 34 items, 0.24% of the values were missing. Normally, it is considered inconsequential when the missing data percentage is below 5% [36]. Therefore, we used an exclude cases listwise method for the descriptive statistics.

In addition to allowing significant items to be retained and interpreted, the emerging factor structure provided insight into the latent factors underpinning the SASQ. The specific approach utilized principal axis factoring (PAF) extraction. An oblique with direct oblimin rotation was selected to improve the interpretation of the data since the factors were likely to be correlated.

The criteria used to determine the number of factors to be extracted included the scree plot [30, 37]. Kaiser's criterion, which advocates for retention of factors with an eigenvalue (EV) $\geq$ 1.0 was applied since low unit values reflect instability in the factor [38]. Further consideration was given to the communalities and factor loadings [34, 37, 39].

**Results.** We followed a combination of approaches that preserve the content validity and psychometric properties of a questionnaire [34, 37–40], and our results indicated that items with high factor loadings were maintained, with the expert views also taken into consideration. It should be noted that we first ran an EFA on the 34 items (S1 Raw data), and due to the exploratory nature of the analysis, a total of 12 items were deleted at this stage, leaving the SASQ with only 22 items. Nine of the items were dropped due to low communalities, below 0.3 (items Q1, Q2, Q4, Q6, Q14, Q15, Q17, Q19, and Q21), and an additional three items (Q20, Q24, Q27) were deleted after in-depth discussions. Therefore, the following results are based on the remaining 22 items (Table 2).

The response distribution in Table 2 indicates that the respondents scored high points on all the items (i.e., *very like me* rather than *very unlike me*) with the lowest score for item 5 (mean: 4.26, SD: 1.38) and the highest score for item 8 (mean: 5.73, SD: 0.97). The skewness and kurtosis coefficients were also examined. Items 11 and 12 did not follow the normal distribution and as such, a Box-cox normality plot was applied to find a transformation that

**Table 2. Descriptive statistics (*N = 543*).**

|  | Mean | SD | Missing value | Kurtosis | Skewness |
|---|---|---|---|---|---|
| Item 1 | 4.34 | 1.23 |  | -1.28 | 1.32 |
| Item 2 | 4.45 | 1.28 |  | -1.64 | 3.59 |
| Item 3 | 5.09 | 1.01 | 2 | 1.39 | 1.15 |
| Item 4 | 4.64 | 1.25 | 1 | 2.04 | 1.56 |
| Item 5 | 4.26 | 1.38 |  | 0.80 | 0.17 |
| Item 6 | 5.15 | 1.16 |  | 1.75 | 1.12 |
| Item 7 | 5.61 | 0.91 | 1 | -1.12 | 2.07 |
| Item 8 | 5.73 | 0.97 |  | 0.85 | 0.16 |
| Item 9 | 4.78 | 1.31 | 2 | -1.87 | 2.00 |
| Item 10 | 4.40 | 1.48 |  | -1.71 | 3.17 |
| Item 11 | 5.40 | 0.95 | 3 | -3.36 | 13.75 |
| Item 12 | 4.85 | 1.21 | 1 | 3.43 | 12.55 |
| Item 13 | 4.74 | 1.35 | 2 | -1.09 | 0.78 |
| Item 14 | 4.95 | 1.20 | 2 | 1.03 | 0.79 |
| Item 15 | 5.29 | 0.91 | 1 | 1.34 | 1.57 |
| Item 16 | 5.24 | 0.91 | 2 | 1.74 | 1.33 |
| Item 17 | 4.92 | 1.41 | 1 | -2.07 | 3.86 |
| Item 18 | 4.95 | 1.33 | 1 | -1.97 | 4.54 |
| Item 19 | 5.00 | 1.21 | 1 | 1.94 | 0.95 |
| Item 20 | 4.79 | 1.19 | 1 | 0.97 | 0.12 |
| Item 21 | 5.08 | 1.11 | 2 | -2.54 | 2.76 |
| Item 22 | 5.21 | 0.91 | 1 | .952 | 0.36 |

approximately normalised the data. This process was then verified by computing the correlation coefficient of a normal probability plot.

The KMO static value was 0.910, which is above the minimum criterion of 0.6. Also, all the KMO values for individual items were greater than 0.77. The Bartlett's test of sphericity had a Chi-square approximation of 4451.76 and was significant at $p < 0.001$. Based on the KMO and Bartlett's sphericity tests, the EFA method was suitable for use in the current research [30, 38, 39].

Studies have suggested that after performing the extraction, the communalities must be greater than 0.30 for good validity to be assumed [30, 38, 39, 41]. The SASQ met most of this criterion, except for items 1 (0.277), 5 (0.261), 10 (0.254), 16 (0.256), and 22 (0.211), which were then eliminated, leaving 17 items. The scree plot revealed that four factors should be retained, consistent with the previous findings [2]. Further, the 22 initial items were grouped into four factors and rotated factor loads varied between 0.310 and 0.843 (Table 3). The four factors explained 64.83% of the variance in the 22-item version.

To determine whether the final four-factor solution was adequate, the determinants of the 22-item correlation matrix and reproduced matrices were used to evaluate model fit [42]. The reproduced correlations were also close to the original correlations, and values in the residual correlation matrix were small, indicating good model fit (Table 4).

### Reliability analysis

**Internal consistency.**    The general reliability static was α = 0.876, indicating good reliability [43, 44]. The correlations between each item and the total questionnaire score were reliable (above 0.3) except for items 5 (0.282) and 22 (0.279), which were finally dropped. The Cronbach alpha values, if deleted, were also calculated and were expected not to exceed 0.876 [31]. This result re-confirmed the deletion of item 5, which had a reliability of 0.877. To ensure that the factor structure had not been affected by deleting item 5, the factor analysis was run again.

## Study 3

The four-factor, 17-item model identified in the EFA was examined using confirmatory factor analysis (CFA), with the intention to configure the final instrument (Fig 1). As a result of this second-order analysis, the hierarchical structure of the instrument and the relationships between variables in the EFA were validated [30, 45]. In addition, acknowledging that predictive validity is one of the most important forms of validity, we similarly evaluated the association between SASQ items using Latent Class Analysis (LCA) method. The combination of these two methodologies constituted an exploratory sequential mixed methodology design approach to integrate social adaptability skills during athletes change from club to club. Similar mixed methodology has been previously applied with success in determining the internal and predictive validity of psychometric instruments [46].

### Confirmatory factor analysis (CFA)

**Participants.**    The participants and data set in study 2 were again used to test the factor structure of the SASQ. The CFA thus used a total of 543 participants (mean age: 16.67, SD: 2.42, range: 12–21 years). We followed the recommendations by DeVellis [35] of participant item ratio of 10:1 in estimating the sample size.

**Data analysis.**    The analysis was conducted using AMOS version 27. The CFA is presented in route diagrams where the circles represent latent variables and the squares represent the observed variables [43–45]. Two-headed arrows indicate covariance between the four latent variables, and single-headed arrows indicate the assumed direction of influence [47]. Within the literature, the most used approximation indices include [34, 48, 49]; (a) the *absolute indices*

**Table 3. Grouping of the 22 items into four factors with rotated factor loads.**

| Item | F1-Coach | F2-Teammates | F3-Family/Friends | F4-Club |
|---|---|---|---|---|
| 8. I always feel stimulated whenever my trainer/coach puts pressure on me to reach my sporting goals. | 0.843 | | | |
| 9. Whenever the coach gives me orders/advice, I am able to listen and respect his/her decision without difficulties. | 0.603 | | | |
| 12. I have difficulties listening to coaches/trainers that I don't know. | 0.585 | | | |
| 19. I wouldn't mind being coached in a foreign language. | 0.541 | | | |
| 3. I can't tolerate it when trainer/coach favours certain players more than others. | 0.536 | | | |
| 17. I find it normal that play time can vary whenever trainers/coaches change | 0.521 | | | |
| 13. I always feel paralysed whenever my trainer/coach puts pressure on me to reach my sporting goals. | 0.461 | | | |
| 7. I feel capable of carrying out my training despite the coach's behaviour (pressure, support, or indifference). | 0.456 | | | |
| 18. In my life, I have always found solutions to satisfy all my sporting and non-sporting interests (performance, studies, friends, family life. . .) | 0.430 | | | |
| 6. I don't support when my coach reduces my play time during matches | 0.386 | | | |
| 11. Whenever I have teammates from a different cultural background, I feel at ease communicating with them. | | 0.691 | | |
| 20. I find that relations between teammates are not usually spontaneous but always interesting. | | 0.553 | | |
| 2. I don't like having foreign teammates. | | 0.467 | | |
| 22. I find it difficult to get recognition from people I don't know well. | | 0.403 | | |
| 10. Whenever I have teammates from different ethnic and cultural backgrounds, I feel don't feel comfortable communicating with them. | | 0.361 | | |
| 15. I don't encounter any particular difficulties whenever I am away from my family (for example, during internships or camps away from home). | | | 0.550 | |
| 14. I don't see myself living far from my family in the future. | | | 0.462 | |
| 5. I always feel lost when I am away from my family (for example, during internships or camps away from my family. | | | 0.319 | |
| 16. I see myself living away from my family in the future | | | 0.310 | |
| 4. I wouldn't accept to play in a club whose training facilities are not of high quality. | | | | 0.385 |
| 21. I know how to train seriously on my own even when the sporting facilities are not satisfactory. | | | | 0.389 |
| 1. I find it acceptable to play at a club regardless of the state of the facilities. | | | | 0.301 |
| **% variance** | **40.78** | **12.98** | **7.83** | **3.24** |

Extraction method: principal axis factoring (PAF)

Rotation method: oblimin with Kaiser normalization

a rotation converged in 15 iterations

referring to the degree to which the covariances implied in the model match the observed covariances through which the free parameters are estimated, and (b) *incremental indices* which refers to the degree to which the model is superior to the alternative model, usually where no covariances between variables are specified- null or independent model.

Despite reporting both the absolute and incremental indices in the current study, the CFA model was validated using the *absolute indices* (i.e., normalized Chi-Square- $X^2$/df and Root Mean Squared Error of Approximation-RMSEA) since an optimal fit is indicated by values close to zero. Normalized Chi-Square ($X^2$) values < 3.0 indicate reasonable adjustment [50, 51] with a value of 5.0 being the minimum acceptable [52]. As concerns RMSEA, values of ≤ 0.06 indicate an adequacy of the model [52–56], but normally the most used cut-off values are [42, 49, 53]; ≤ 0.05 good fit, ≤ 0.08 acceptable fit, ≤ 0.10 indicate a mediocre fit, and > 0.10 a poor (un acceptable fit). Factor weights (FL; factor loading) were also computed

**Table 4. Reproduced correlation matrix.**

|        | I-1   | I-2   | I-3   | I-4   | I-5   | I-6   | I-7   | I-8   | I-9   | I-10  | I-11  | I-12  | I-13  | I-14  | I-15  | I-16  | I-17  | I-18  | I-19  | I-20  | I-21  | I22   |
|--------|-------|-------|-------|-------|-------|-------|-------|-------|-------|-------|-------|-------|-------|-------|-------|-------|-------|-------|-------|-------|-------|-------|
| I-1  | .277a |       |       |       |       |       |       |       |       |       |       |       |       |       |       |       |       |       |       |       |       |       |
| I-2  | .254  | .338a |       |       |       |       |       |       |       |       |       |       |       |       |       |       |       |       |       |       |       |       |
| I-3  | .283  | .247  | .351a |       |       |       |       |       |       |       |       |       |       |       |       |       |       |       |       |       |       |       |
| I-4  | .304  | .209  | .34   | .386a |       |       |       |       |       |       |       |       |       |       |       |       |       |       |       |       |       |       |
| I-5  | .177  | .107  | .124  | .207  | .261a |       |       |       |       |       |       |       |       |       |       |       |       |       |       |       |       |       |
| I-6  | .318  | .353  | .335  | .315  | .169  | .409a |       |       |       |       |       |       |       |       |       |       |       |       |       |       |       |       |
| I-7  | .242  | .356  | .291  | .199  | .029  | .374  | .440a |       |       |       |       |       |       |       |       |       |       |       |       |       |       |       |
| I-8  | .305  | .314  | .444  | .361  | .028  | .406  | .442  | .654a |       |       |       |       |       |       |       |       |       |       |       |       |       |       |
| I-9  | .247  | .221  | .343  | .306  | .105  | .316  | .302  | .469  | .386a |       |       |       |       |       |       |       |       |       |       |       |       |       |
| I-10 | .191  | .246  | .163  | .153  | .168  | .270  | .241  | .175  | .177  | .254a |       |       |       |       |       |       |       |       |       |       |       |       |
| I-11 | .241  | .348  | .155  | .153  | .193  | .336  | .300  | .132  | .113  | .308  | .470a |       |       |       |       |       |       |       |       |       |       |       |
| I-12 | .237  | .192  | .334  | .307  | .144  | .302  | .271  | .445  | .402  | .196  | .098  | .445a |       |       |       |       |       |       |       |       |       |       |
| I-13 | .183  | .271  | .252  | .160  | .038  | .306  | .373  | .403  | .317  | .230  | .212  | .327  | .383a |       |       |       |       |       |       |       |       |       |
| I-14 | .179  | .135  | .144  | .199  | .266  | .202  | .090  | .083  | .169  | .224  | .218  | .229  | .136  | .316a |       |       |       |       |       |       |       |       |
| I-15 | .143  | .136  | .119  | .142  | .236  | .193  | .121  | .088  | .182  | .248  | .220  | .256  | .198  | .323  | .367a |       |       |       |       |       |       |       |
| I-16 | .200  | .199  | .210  | .210  | .186  | .261  | .212  | .235  | .246  | .230  | .219  | .285  | .239  | .252  | .275  | .256a |       |       |       |       |       |       |
| I-17 | .290  | .221  | .359  | .367  | .241  | .348  | .264  | .425  | .419  | .224  | .169  | .478  | .323  | .325  | .341  | .345  | .547a |       |       |       |       |       |
| I-18 | .290  | .343  | .324  | .280  | .120  | .391  | .389  | .426  | .320  | .248  | .305  | .303  | .329  | .164  | .169  | .245  | .329  | .386a |       |       |       |       |
| I-19 | .287  | .289  | .354  | .320  | .131  | .370  | .349  | .465  | .373  | .230  | .213  | .375  | .328  | .185  | .190  | .261  | .403  | .365  | .388a |       |       |       |
| I-20 | .208  | .295  | .144  | .139  | .172  | .293  | .263  | .133  | .122  | .272  | .393  | .118  | .203  | .205  | .215  | .208  | .180  | .268  | .200  | .333a |       |       |
| I-21 | .243  | .204  | .294  | .293  | .209  | .305  | .242  | .350  | .352  | .240  | .176  | .406  | .297  | .292  | .316  | .308  | .467  | .290  | .344  | .182  | .406a |       |
| I-22 | .134  | .191  | .071  | .080  | .160  | .191  | .154  | .034  | .067  | .215  | .295  | .082  | .134  | .195  | .210  | .170  | .143  | .168  | .121  | .254  | .148  | .211a |

**I** = Item

a) Reproduced communalities

b) Residuals are computed between observed and reproduced correlations. There are 21 (9.0%) nonredundant residuals with absolute values > 0.05

with values equal to or greater ($\geq 0.5$) being accepted [30]. Ideally, according to Hair [38] accepted values should be over 0.7.

**Results.** The Chi-square value ($X^2$) was 327.96 ($p = 0.000$) (Table 5). However, it has been argued that $X^2$ static value shows high sensitivity to larger sample sizes like ours [47]. Therefore, a normalized Chi-Square ($X^2/df$) value was reported since it reduced the test sensitivity to the sample size and model complexity. In the current study, the RMSEA value was 90% CI < 0.06, indicating model adequacy.

As concerns the incremental fit, the comparative fit index (CFI), the Tucker-Lewis index (TLI), and the goodness-of-fit index (GFI) values indicated a good fit to the model (CFI = 0.809, TLI = 0.844, and GFI = 0.926, respectively). The CFI, TLI, and GFI have a range of 0 to 1, with values indicating greater validity as they approach 1 [30].

Last, the latent variables for the four subscales of the SAS had nine, three, three, and two items each (Fig 1). According to the route diagram, all items show acceptable factor loads (standardized values) ranging from 0.66 to 2.74, except for teammates factor (Q12R) which had a value of 0.31. Nine elements of the coach construct had factorial loads between 1.00 and 2.74, as did the three elements of the teammates construct, which had factorial loads between 0.31 and 1.00. The three elements of the family construct ranged from 0.95 to 1.13, and the last two elements of the club construct ranged from 0.66 to 1.00.

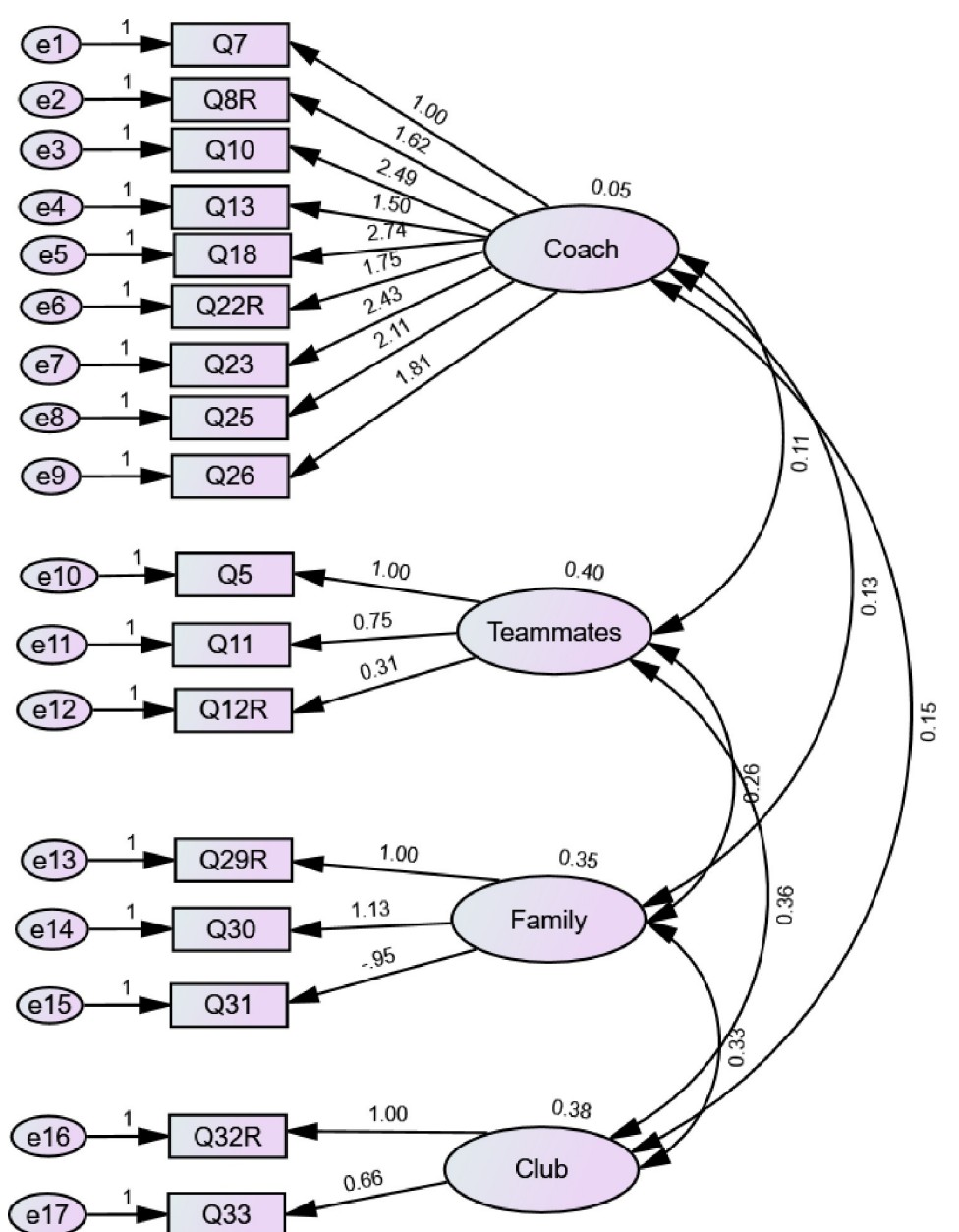

**Fig 1. Confirmatory factor analysis diagram.**

**Table 5. Fit test of the Chi-square adjustment test.**

| Model | NPAR | CMIN | DF | P | CMIN/DF |
|---|---|---|---|---|---|
| Default model | 42 | 327.960 | 111 | 0.000 | 2.955 |
| Saturated model | 153 | 0.000 | 0 | | |
| Independence model | 17 | 1528.595 | 136 | 0.000 | 11.240 |

## Latent class analysis

**Participants and data analysis.** The same data of 543 participants used in CFA was reused to classify the Social Adaptability Skills (SAS) into meaningful groups using latent class analysis (LCA) [57]. According to Nylund, Asparouhov and Muthen [58], LCA models are used to, "uncover unobserved heterogeneity in a population and to find substantively meaningful groups of people that are similar in their responses to measured variables" (p. 536). The current study settled on LCA because it transcends other conventional parametric analyses by relaxing the assumptions of homogeneity of variance, skewness, normal distribution, and linearity that cannot be analysed using the same conventional parametric clustering techniques [59]. The analysis was performed on Latent Gold® 6.0 [59] with the maximum likelihood method for parameter estimation. The LCA in the present study comprised: (a) building and classification of a cluster model (b) bootstrapping to confirm optimum fit model, and (c) predicting the precision of the optimal model using classification statistics.

The optimal model was determined by comparing the fit statistics of three models [60] namely: (i) Akaike Information Criterion (AIC): a measure of relative fit or quality of the model of the data, which considers model parsimony (the number of parameters). AIC is estimated based on log-likelihood squared ($L^2$) and LL. In either case, lower AIC indices indicate a better fit [61], (ii) Consistent AIC (CAIC) based on LL, and (iii) Bayesian Information Criterion (BIC) based on LL: A better fit is indicated by low BIC indices when comparing different models. Furthermore, BIC imposes a stronger penalty on the number of parameters than AIC. Therefore, when AIC and other fit statistics differ, the former is usually prioritized when assessing model fit [61].

It should be noted that, during the analysis, the competing models were nested with the optimal model in order to preserve the same number of latent classes and variables on each model. An examination of the statistical significance ($p$ value) of the conditional bootstrapping analysis determined the amount of improvement in fit. If the $p$ value generated in conditional bootstrapping is statistically significant ($p < 0.5$), then the competing model provides a statistically significant improvement in model fit, even though it might compromise inferior AIC, BIC, and CAIC [62, 63]. Finally, in estimating the effect size, an entropy $R^2$ was computed using the sample size, posterior probabilities, and a number of classes [63]. According to Kaplan and Keller [64], high entropy $R^2$ values indicate high precision in classification and the prediction of the emerging latent classes.

## Results

**Latent class analysis.** Fig 2 illustrates conditional probabilities of the three classes on a profile graph. On the graph, four significant variables are displayed on the horizontal axis against probability values on the vertical axis, making it easier to observe and interpret the comparison between latent classes. From the graph, class three obtained the highest rating on each corresponding dimensions (coach, teammates, family, and club) maximum score of six on all items. Class 1 obtained the lowest rating on each corresponding dimension's score while class two obtained average ratings an each corresponding dimension.

Following the LCA analysis, the SASQ 17- items that significantly contributed to discriminating participants were determined.. Table 6 demonstrates the loadings indicating "how well" the indicator is explained by the model [63, p. 113], and mean scores of the item across the three groups, alongside their $p$ values estimated based on Wald statistics. The loading indices are the magnitude of the contribution of the variable to measuring the construct. All items had higher loadings ($> 0.5$) except for item Q22R which reported a loading of 0.489. The three groups of participants were differentiated based on the SASQ 17-item questions. For example,

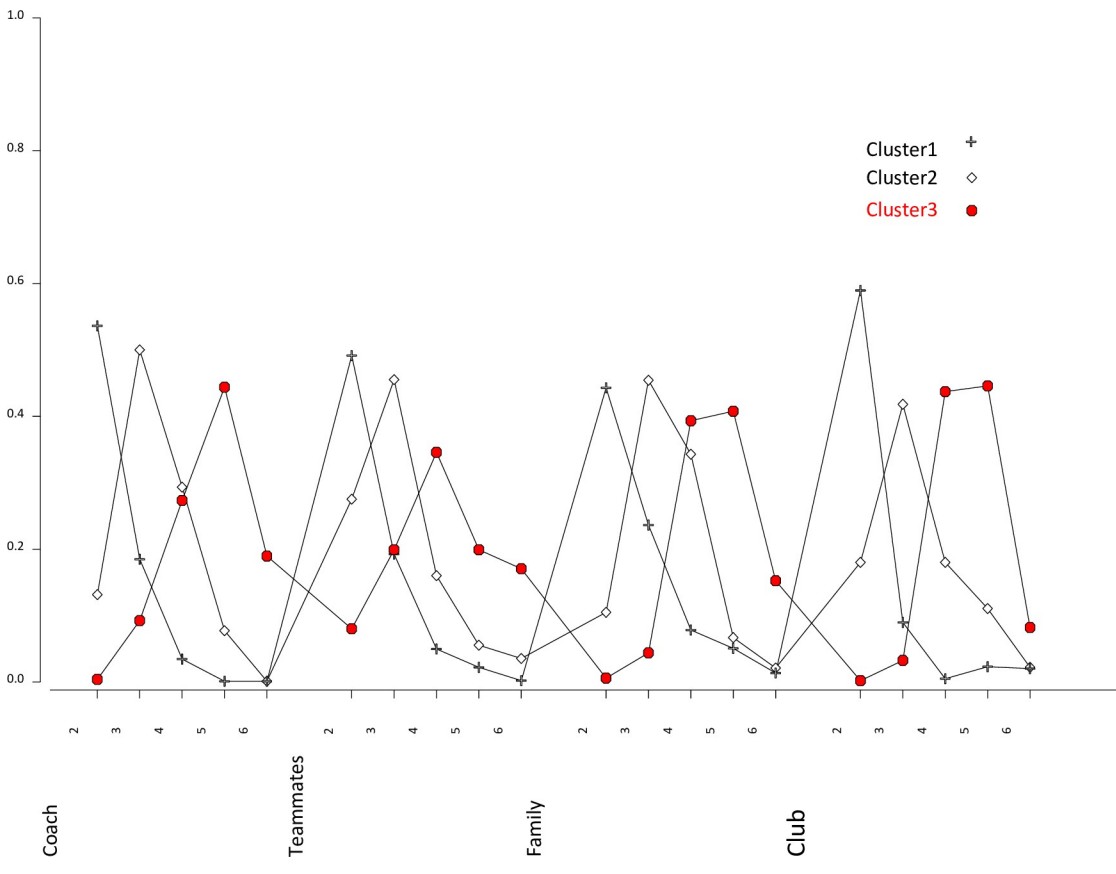

**Fig 2. Profile plot for the three-class model.**

high achievers had the highest mean (75.047) for item Q5 followed by average achievers (54.711) and finally low achievers (53.655), meaning that the response of high achievers as regards item Q5 was significantly higher than responses generated by participants in average and low achievers group. Similarly, response concerning item Q25 reported that high achievers ($M$ = 17.229) were significantly higher than those of average and low achievers groups ($M_{1,2}$ = 12.339, 11.119) ($p < 0.05$).

Four models were fitted to the data and as demonstrated in Table 7, a three class model differentiating between three different groups (low achievers, average achievers, and high achievers) had an optimal fit to the data (BIC = 4172.964; AIC = 3437.369; CAIC = 3642.426). Although the model had a higher AIC index than that of a the four-class model, however, since it had lower BIC and CAIC, a smaller number of parameters to estimate (161; more parsimonious), and a significant conditional bootstrapping $p$ value, there was evidence for its optimal fit. Finally, the effect sizes computed by the entropy $R^2$ index were 0.8658 (two class), 0.9184 (three class), and 0.9738 (four class), which translated into 87%, 91%, and 97% respectively of accuracy in classifying the participants into three latent classes and the prediction of their SASQ test scores.

**Internal reliability.** Once the CFA and LTA validated the SASQ factorial structure with 17 items (see S1 Appendix), a reliability analysis to measure the internal consistency was performed using Cronbach's alpha (α). Nevertheless, its limitations are well known [65], including the assumptions of uncorrelated errors, tau-equivalence and normality. Therefore, the current study analysed the internal consistency using composite reliability, which provided a

**Table 6. Classification of the SASQ items in three class model.**

| Item | Loading | Wald | High Achievers (*Mean*) | Average Achievers (*Mean*) | Low Achievers (*Mean*) | *p* value |
|---|---|---|---|---|---|---|
| Q5 | 0.563 | 72.452 | **75.047** | 54.711 | 53.655 | *p* < .05 |
| Q11 | 0.501 | 53.404 | 5.352 | 11.630 | **13.772** | *p* < .05 |
| Q12R | 0.575 | 59.789 | **12.300** | 8.433 | 11.758 | *p* < .05 |
| Q7 | 0.703 | 69.770 | 18.784 | **19.031** | 11.643 | *p* < .05 |
| Q8R | 0.582 | 59.642 | **12.571** | 12.423 | 10.756 | *p* < .05 |
| Q10 | 0.511 | 50.489 | **16.136** | 7.248 | 8.606 | *p* < .05 |
| Q13 | 0.609 | 59.441 | **12.799** | 5.122 | 12.375 | *p* < .05 |
| Q18 | 0.526 | 61.481 | 13.436 | 8.845 | **13.491** | *p* < .05 |
| Q22R | 0.489* | 82.751 | 19.379 | 8.800 | **22.827** | *p* >.05* |
| Q23 | 0.715 | 66.203 | **16.354** | 8.680 | 16.350 | *p* < .05 |
| Q25 | 0.612 | 65.467 | **17.229** | 12.339 | 11.119 | *p* < .05 |
| Q26 | 0.506 | 55.277 | **17.202** | 12.031 | 8.116 | *p* < .05 |
| Q29R | 0.502 | 52.191 | 12.405 | 12.538 | **13.538** | *p* < .05 |
| Q30 | 0.519 | 63.891 | **16.047** | 2.261 | 10.876 | *p* < .05 |
| Q31 | 0.513 | 53.476 | 6.904 | 7.162 | **13.104** | *p* < .05 |
| Q32R | 0.531 | 50.573 | 9.224 | **11.697** | 5.966 | *p* < .05 |
| Q33 | 0.509 | 56.970 | 11.315 | 11.968 | **13.245** | *p* < .05 |

*Note*: Areas marked (*) indicates non-significance

Bold print indicate highest mean scores.

better option (Table 8). Raykov's formula was used to calculate the composite reliability [66]. An acceptable threshold for composite reliability falls between 0.6 to 0.7 but should not be above 0.95 [38]. This means that the underlying SASQ factors presented acceptable internal consistency and can be used reliably in future research or real-world applications.

**Test re-test reliability.** In order to evaluate the reliability, a test-retest analysis was carried out [30, 67–69], with the same subject measured at two different times (always in conditions of similar application) [33, 67, 69].

**Participants.** One hundred and seventy-four athletes (males *N* = 94 and females *N* = 80) performing their sport at the regional (*N* = 120) or national (*N* = 54) level participated in this study. The mean age was 17 years (age range: 12–21; SD = 1.79). All participated in team sports (football *N* = 48, 27.6%; basketball *N* = 67, 38.5%; volleyball *N* = 25, 14.4%; rugby *N* = 10, 5.7%; and unihockey *N* = 24, 13.8%). Initially, 189 participants took part in the first administration of the SASQ; however, only 174 participants participated in both administrations. Since we

**Table 7. Comparative fit statistics of the LCA models.**

| Model-Class | BIC | AIC | CAIC | Npar | L² | df | Entropy R² | R² | *Pcond.Boot* |
|---|---|---|---|---|---|---|---|---|---|
| One | 6790.451 | 6524.145 | 6586.145 | 62 | 656.1892 | 480 | 0.7706 | | |
| Two | 5753.671 | 4327.642 | 4659.642 | 332 | 3566.915 | 210 | 0.8908 | 0.8658 | |
| Three | **4172.964** | 3481.426 | **3642.426** | 161 | 3159.426 | 381 | 1 | 0.9184 | *p* < .05 |
| Four | 4824.74 | **3437.369** | 3760.369 | 323 | 2791.369 | 219 | 0.7671 | 0.9738 | *p* > .05 |

*Note*: BIC = Bayesian Information Criterion; AIC = Akaike Information Criterion

CAIC = Consistent Akaike Information Criterion

Bold print indicate the lowest fit index.

**Table 8. Composite reliability of the four factors.**

|  | Teammates | Coach | Family | Club |
|---|---|---|---|---|
| Composite reliability value | 0.76 | 0.76 | 0.72 | 0.68 |

had 17 items, the sample size ($N$ = 174) met the empirical rule of the 10:1 participant-items ratio recommended in the literature [30, 35].

**Procedure.** An approval was granted by the Lausanne University Research Ethics Commission (Project number: E_SSP_122021_00002) for the study. The procedures were explained to participants, and they subsequently gave their written consent, along with their parents or guardians if they were under 16 years of age. Between August and October 2022, paper-pencil SASQ questionnaires were distributed to the participants. At the first administration, participants were assigned a code name based on their initials, age, and the initials of their parents (mother and father). Scale reliability was assessed by completing the questionnaire a second time, with code names to ensure correspondence between results at time 1 ($T_1$) and time 2 ($T_2$). According to Bonnet's guidelines [70] the sample size to obtain a 95% CI for the intra-class correlation coefficient (ICC) ($p$) with a desired width of 0.2 for two repetitions is greater than 159 if $p$ is $\geq$ 0.6. The current study recruited 174 participants which was well above the recommendation. The SASQ instrument was administered twice at 4-week intervals.

**Data analysis.** Descriptive statistics were performed on the 17 items with means, standard deviations, measures of skewness and kurtosis, and analysis of missing values. In the test-retest reliability analysis, we used the ICC (two-way mixed effects, absolute agreement) and a paired sample $t$-test to compare the factors between measurement times. Koo and colleague [44], defined ICC values less than 0.5 indicated low reliability; between 0.5 and 0.75, moderate; between 0.75 and 0.90, good; and above 0.90, excellent. All data were analysed using SPSS (version 27).

**Results.** Scale reliability was supported by test-retest data (Table 9) (S1 Dataset).

In general, participants presented moderate to high scores on the factors at $T_1$ (mean: 4.14–5.17) and $T_2$ (mean: 4.13–5.33), with corresponding similar standard deviations: $T_1$ (SD = 0.15–0.79) and $T_2$ SD = 0.06–0.76, respectively. According to the ICC scores, the coach factor showed "excellent reliability" between $T_1$ and $T_2$ and teammates and family factors showed "good reliability". However, the ICC scores for the club factor indicated "moderate reliability". Cronbach's alpha varied minimally between $T_1$ and $T_2$ on all the four factors. A paired $t$-test indicated no significant difference between the two administrations.

## Discriminant validity analysis

According to Hair, discriminant validity ensures that a construct measure is empirically unique and represents phenomena of interest that other measures in a structural equation model do not capture [71]. Essentially, discriminant validity requires that test results are not

**Table 9. Descriptive statistics, internal reliability and intraclass correlation ($N$ = 174).**

|  | $T_1$ | $T_2$ | Paired $t$-test |  | $T_1$ | $T_2$ | ICC |
|---|---|---|---|---|---|---|---|
|  | M(SD) | M(SD) | t(173) | p | α | α |  |
| Coach | 4.14 (.79) | 4.13(.76) | 0.168 | 0.871 | 0.910 | 0.940 | 0.983 |
| Teammates | 5.17(.50) | 5.19(.72) | -.089 | 0.937 | 0.890 | 0.910 | 0.842 |
| Family | 4.73(.26) | 4.90(.15) | -2.350 | 0.143 | 0.750 | 0.790 | 0.786 |
| Club | 4.99(.15) | 5.33(.06) | -5.154 | 0.122 | 0.680 | 0.710 | 0.606 |

highly correlated with measures from which they are supposed to differ [72]. In the current study, we propose the heterotrait-monotrait ratio of correlations (HTMT) as a new approach to assess discriminant validity in variance-based SEM [73]. This is due to the fact that recent research suggests that the routinely used Fornell-Larcker criterion is not effective under certain circumstances of discriminant validity testing [74].

Since there are two ways of using the HTMT to assess discriminant validity: (i) as a criterion or (ii) as a statistical test, the former involves comparing it to a predefined threshold with the exact threshold levels of the HTMT being debatable [43, 75]. We therefore chose to use the statistical test to assess the HTMT. This was carried out by applying a bootstrapping procedure to construct confidence intervals for the HTMT in order to test the null hypothesis ($H_0$: HTMT $\geq$ 1) against the alternative hypothesis ($H_1$: HTMT $<$ 1). A confidence interval (CI) containing the value one (i.e., $H_0$ holds) indicates a lack of discriminant validity. Conversely, if the value one falls outside the interval's range, this suggests that the two constructs are empirically distinct. The advantage of using CI is that they provide more information by highlighting the direction and magnitude of the difference, or, if the hypothesis is not rejected, the power of the procedure can be assessed by the width of the interval [76].

**Results.**   The computations yields values between (0.661 and 0.849) (Table 10). Upon comparing if the CI result value (one) falls outside the interval range, it can be reported that all the four constructs are empirically distinct.

**Convergent validity.**   In order to confirm the extent to which the measures capture the common SASQ constructs, a convergent validity analysis was performed. It involved calculating the average factor loading for all the measures in each SASQ construct (i.e., the sum of the squared loadings divided by the number of measures). Convergent validities above $r = .70$ are recommended, whereas those below $r = .50$ should be avoided [77]. However, research evidence suggests that actual levels of convergent validity in psychology research still vary widely [77]. Therefore, without more specific guidance, researchers reach logically inconsistent conclusions, arguing that a convergent validity as low as $r = .28$ [e.g.,78] indicates the measures converge, whereas others report convergent analysis as high as $r = .75$ [e.g., 79] signalling high convergence.

**Results.**   Results in Table 11 indicate that the measures of coach and teammates constructs (0.732 and 0.761) respectively reported high convergent validity. In contrast, both the measures of family and club constructs (0.657 and 0.631) respectively reported average or modest convergent validity.

## Study 4

In this fourth study, we standardized the SASQ values by generating scores that ranked the measured factors in order to increase the reliability and generalizability.

**Table 10. HTMT results.**

|           | Coach | Teammates | Family | Club |
|-----------|-------|-----------|--------|------|
| **Coach** |       |           |        |      |
| **Teammates** | .733 |        |        |      |
|           | CI.900 (0.713; 0.754) |  |  |  |
| **Family** | .676 | .849 |        |      |
|           | CI.900 (0.653; 0.699) | CI.900 (0.803; 0.895) |  |  |
| **Club**  | .742 | .766 | .661 |      |
|           | CI.900 (0.719; 0.765) | CI.900 (0.748; 0.784) | CI.900 (0.633; 0.689) |  |

**Table 11. Convergent validity results.**

| Measures | Average factor loading (*r*) |
| --- | --- |
| Coach | **0.732** |
| Teammates | **0.761** |
| Family | 0.657 |
| Club | 0.631 |

## Participants

The participants and dataset in study 2 were again used to standardize the SASQ scores. The standardization analysis thus used 543 participants (mean age: 16.67, SD: 2.42, range: 12–21 years).

**Method.**   Once descriptive analyses were performed on the raw scores, focus was put on standardizing the SASQ scores. This was done using the *R* statistical software package, with the raw scores transformed into T-scores to calculate the median scores for each of the four SASQ dimensions (S2 Dataset). Once the median score was established, three distribution groups were formed: low achievers, average achievers, and high achievers (Fig 3).

Scores ≤ 40% designated low achievers and scores ≥ 60% designated high achievers. For the average achievers, a score was established by determining the average between the low and high achievers, resulting in a 50% score representation. For ease of interpretation, high achievers were described as athletes who were able to deploy the necessary SAS to deal with problematic experiences during club changes. In contrast, low achievers were described as athletes who did not have these skills. Average achievers were those athletes who had some but not all of these SAS.

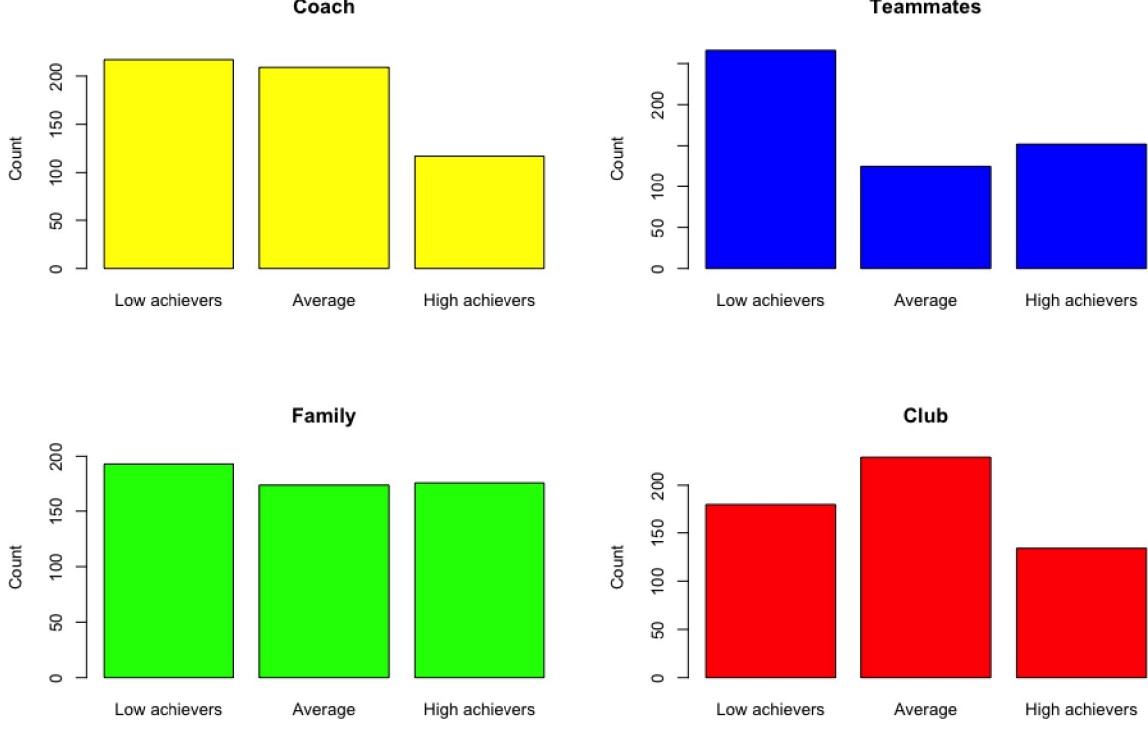

**Fig 3. Group distribution/ranks of each of the four SASQ dimensions.**

**Table 12. SASQ scoring guide.**

| Dimension (max score) | Low achievers ≤ 40% | Average achievers 41% to 59% | High achievers ≥ 60% |
|---|---|---|---|
| Coach (54) | ≤ 22 | 23 to 31 | ≥ 32 |
| Teammates (18) | ≤ 7 | 8 to 10 | ≥ 11 |
| Family (18) | ≤ 7 | 8 to 10 | ≥ 11 |
| Club (12) | ≤ 5 | 6 | ≥ 7 |

**Results.** For the coach dimension, most of the scores indicated low achievers, with 40% of the total score. In contrast, average and high achievers respectively represented 38% and 22%. In general, these results indicated that most of the athletes encountered problematic situations concerning the coaching dimension. For the teammates dimension, the low achiever group represented the highest percentage of the total score with 49%, followed by high achievers at 28% and average achievers at 23%. The club dimension indicated that average achievers were the majority, with 42% of the total score as opposed to 33% and 25% for the low and high achievers, respectively. A different situation was observed for the family dimension in that low achievers represented 36% of the total score, while the average and high achievers each represented 32%. In general, apart from the club dimension, for which the average achievers were the majority, low achievers were the majority in the remaining three dimensions, clearly indicating that within the general French-speaking Swiss youth athlete population, most do not possess or deploy the SAS skills during problematic club changes.

*An example.* Alex, a footballer, obtained the following scores: coach: 20, teammates: 15, family: 9, and club:10, for a total SASQ score of 54. Using Table 12, Alex is a low achiever for the score on the coach dimension, part of 40% of the general French-speaking Swiss population. Thus, 60% of this general population had higher scores than Alex. The scores on the teammates dimension indicate that he is a high achiever and has higher scores than 59% of our general study population. The score of 9 on the family dimension presents Alex as an average achiever, having outscored 40% of our general study population while another 40% had higher scores than he did. Last, with a score of 10 on the club dimension, he was a high achiever, with a higher score than 59% of our general study population. Overall, Alex's total percentage SASQ score was 53%.

An examination of each of the three group means found that most individuals rated on SASQ were classified as low achievers on all of the three dimensions (i.e., coach teammates, and family) except on club dimension which was represented mostly by average achievers (Table 13). The results above indicate that, in general, most participants lacked the possession and deployment of SAS and that only a few individuals possessed and deployed the necessary SAS.

**Table 13. SASQ distribution profile as per dimension.**

| Dimension | Low Achievers | Average Achievers | High Achievers | Total SASQ Score per dimension |
|---|---|---|---|---|
| Coach | **21.6** | 20.52 | 11.88 | 54 |
| Teammates | **8.82** | 4.14 | 5.04 | 18 |
| Club | 3.96 | **5.04** | 3.00 | 12 |
| Family | **6.48** | 5.76 | 5.76 | 18 |

*Note*: Bold print indicate highest mean scores

## Discussion

The aim of the current study was to outline the initial development and validation of a questionnaire designed to assess the SAS levels of young athletes in talent development. In addition to providing regular feedback and refinement, SAQ instrument was also expected to identify areas that needed immediate attention, improvement or maintenance.

The current study demonstrated acceptable internal consistency when compared with previous studies and reference instruments, such as the Psychological Characteristics of Developing Excellence (PCDEQ) questionnaire [4] or the Talent Development Environment Questionnaire (TDEQ) [5]. More specifically, given that the data from EFA analysis presented a general reliability of $\alpha = 0.876$ and the reliability for each of item ranging from $\alpha = 0.865$ to $\alpha = 0.875$, SASQ should be considered a reliable instrument. Compared to the reliability scores of other instruments that measure social skills, example: the Social Skills Questionnaire (CHASO III) [13] with $\alpha = 0.70$, the Social Skills Questionnaire for College Students (SSQ-U) [14] with $\alpha = 0.70$, or the Social Skills Questionnaire for Traumatic Brain Injury (SSQ-TBI) [80] with $\alpha = 0.80$. The SASQ provides reliability in that, it can measure the situated social skills that athletes possess and, therefore, it can be relied upon by researchers.

According to previous research [2], the 17 items spread over four factors were also satisfactory. Since the objective of the current study was to develop and validate an easily understandable instrument that measures relevant dimensions of the problematic experiences encountered by athletes during club changes, a 17 item instrument was deemed important. The fact that the SAQ instrument retained the previously proposed four-factor model increases its value. Compared to long, complicated scales that have been applied in previous studies (e.g., TDEQ with 68 items [5], Social Skills Questionnaire (CHASO III) with 76 items [13], Social Skills Inventory with 90 items [81], and the Social Skills Questionnaire for College Students (SSQ-U) with 38 items [14], shorter psychometric instruments are preferable [82]).

The structural equation modelling (SEM) results indicated that, in general, the SASQ model was adequate for validating the constructs. The EFA enabled us to reduce the number of SASQ items from 34 to 17 with four dimensions by selecting items that represented better psychometric properties. Doing so resulted in adequate levels of correlation, which indicates excellent validity, as confirmed through CFA yielding four stable factors. In addition, the construct validity of the SASQ agreed with the findings of Owiti and Hauw [2], who also reported a four-factor model related to coach, teammates, family and friends, and club. In this sense, the dimensions of the resulting scale encompass all possible levels of the social adaptability skills that athletes require during club changes.

Regarding the model fit of the CFA, we considered various criteria in validating the fit indices. According to various research [30, 31, 54, 55, 83], RMSEA values greater than 0.1 indicate poor performance, between 0.08 and 0.10, acceptable, and less than 0.05 indicate acceptable performance. The current study reported a value of 0.06 indicating an acceptable fit indices. The TLI value of 0.844 in the current study was a little below the criterion since an acceptable value should be over 0.9 for a good fit [50–52, 54–56]. The Chi-square result was significant, and it has been suggested that this value is sensitive to the sample size. It was therefore normalized and had an acceptable result of below 3 [47, 48, 84]. Based on these indices, this sample fits a four-factor model reasonably well.

Accordingly, in terms of factor loading, factor weights are usually considered significant when equal to or greater than 0.5 (FL $\geq 0.50$) [31]. However, some authors suggest that 0.30 is the minimum requirement for interpreting a sample size $\geq 350$ [34, 36, 38]. The CFA revealed that 15 loadings were greater than 0.30 (range: 0.30 to 0.62) and only two loadings were below 0.30 (range: 0.26 to 0.29). Low factor loadings suggest that the items are correlated, thus not

measuring well-separated latent concepts. We offer two possible explanations as regards the low factor loadings; first, in a real world setting, it is extremely difficult to find latent factors that make up one concept being absolutely independent. As for the case of SASQ which measures the problematic experiences of players during club changes, these experiences may be closely associated and therefore expressed in the same way by the participants. Second, since the participants formed a homogeneous group, as all were aspiring athletes, it is possible that the distinguishing experiences of a specific difficulty were not expressed.

The computational yield for the discriminant validity in the current study reported the following results: (Coach and Teammates = 0.733; Coach and Family = 0.676; Coach and Club = 0.742; Family and Teammates = 0.849; Club and Family = 0.661; Teammates and Club = 0.766). Comparing these results with the threshold values as defined in HTMT [77], suggested that discriminant validity had been established. Further test of convergent validity which reflects the extent to which measures capture common constructs reported the following results: (Coach = 0.732; Teammates = 0.761; Family = 0.657; Club = 0.631). Convergent validity range ($r$ = .70 and above) being recommended whereas range ($r$ = 0.5 and below) is to be rejected [77]. Despite the favourable convergent validity results reported in the current study, research evidence suggest that actual levels of convergent validity still vary widely [77].

It should be noted that good practice dictates a minimum of three items per factor [52, 85]. Three of the SASQ factors consisted of at least three items per factor, although the club factor had only two items. In retaining the club factor with only two items, we present the following arguments: First, there was a strong theoretical and practical reason linked to the observation that future athletes are already self-motivated to pursue their careers despite problematic challenges concerning the club. Second, our decision is supported by previous research in the psychology literature suggesting that constructs that do not have a wide domain or those that are not conceptualized as multidimensional may present single-item measures [86–89]. Third, a factor with two items is only considered reliable when the items are highly correlated ($r$ = 0.70) but fairly uncorrelated with the other items, as was the case in the current study [90].

Concerns have been raised about considering psychometric measures as non-interdependent scales because the reliability of single-item scales can be legitimately questioned [91]. This suggestion extends to the SASQ-17 item psychometric measure which cannot be considered as uncorrelated [92]. However, since validity implies reliability [93, 94] the choice of the most relevant SASQ items were based on pragmatic grounds including their predictive value for relevant outcomes. It is therefore essential to emphasize the importance of external validity when choosing the most suitable SASQ scores, some of which should be correlated or not with each other. Adequate SASQ scores should not be selected to internal validity evidence alone [46].

The results of the LCA for predicting SAS disposition and deployment by athletes replicated findings of the CFA underlying the benefits of combining two methodologies. In applying the predictive validity of the SASQ items, we confirmed the existence of three specific participant profiles that could be differentially related to SAS during club to club change. Results of the LCA suggested that the SASQ could discriminate participants into three characteristic profiles (example: low achievers, average achievers, and high achievers). In line with the LCA approach, athletes could be clustered into meaningful and relatively homogenous classes on the basis of the SASQ-17 item scores.

According to our findings, categories established on the basis of the SASQ 17-items were associated with distinct outcomes such as a participant being either in a low, average, or high achiever group. This highlighted the fact that it was possible to perform SASQ scores based predictions through this process of group categorization. In this case SASQ data could be collected as part of routine psychosocial skills training within talent and development programs. Indeed and in line with the results of the present study, SASQ items could be used as a first

step to identify specific subgroups when athletes change from club to club. In a second step, specific profile membership and its association with important variables could be used to inform the coach and club decisions.

The LCA also reported optimal fit for SASQ items based on BIC, AIC, and CAIC results, this was a confirmation of similar results of "goodness fit" reported in CFA. However, it should be noted that the LCA reported a non-significant result on item Q22R (*I always feel paralysed whenever my trainer/coach puts pressure on me to reach my sporting goals)* on its inability to discriminate participants as concerns difficulties encountered with coaches. However, this was not the case with the CFA model.

At this point, we have to admit that despite equipping young athletes with these psychosocial skills, it does not necessarily guarantee that they will reach elite level competition, since there exists other variables that could influence the possibility of achieving success. However, their absence may prevent these athletes in overcoming obstacles along their success pathway [3, 10].

## Limitations

Although we developed a valid and reliable instrument to measure social adaptability skills during club changes, our work has some limitations that must be kept in mind when using the instrument. First, the measure was constructed based on data from six team sports, which makes generalizability a limitation. Therefore, before using SASQ in other sports, evidence of the validity and reliability of the instrument in those sports is needed. In addition, our sample was drawn from the French-speaking part of Switzerland, which points to the culture-based approach of our standardization of the SASQ. Further research might focus on multicultural validation of the SASQ.

Second, considering that the participants were nested within teams, multilevel analysis could have also been applied. The this type of analysis might have provided more accurate parameter estimates, standard errors, and associated tests of significance, as well as allowing us to examine variance by sport type. In future, researchers should use multilevel modelling to examine SASQ data and also determine whether there are systematic differences between sport types and the four factors.

Third, due to the general interest, we have to admit that the LCA reported Q22R as having low factor loading, however, the CFA confirmed the same Q22R as having contributed to the factor structure of SASQ.

## Implication for practice

According to the current findings, although new, SASQ instrument appears to be useful for researchers, coaches, and clubs alike. Individuals working with athletes can use the instrument to predict whether young athletes have the necessary social adaptability skills to handle club changes and, based on the results, they can design intervention and inclusion programs for them. During such interventions, the constructs of the SASQ can be measured before and after the club change in order to evaluate any changes in these behavioural precursors.

For the young athletes to develop the SAS necessary to overcome future obstacles, their immediate environment needs to be carefully managed. It is also important to integrate education programs designed to evaluate and develop these SAS, whether they take the form of formal or informal psychosocial training within talent identification and development programs.

Professional players may also benefit from the findings of the current study. In order to facilitate smooth transitions between clubs, they could be encouraged to assess whether they possess the necessary skills. SASQ has the potential to benefit coaches by providing insight that can enhance players' transition from one club to another and focusing on development.

Professional sports clubs and high-performance organizations could also use the questionnaire for identifying major adaptation barriers faced by incoming players. Last, the findings can also be used by sports clubs to maintain and enhance existing resources that help players with transitions.

## Supporting information

**S1 Raw data.**
(XLSX)

**S1 Appendix. SASQ 17-item questionnaire.**
(DOCX)

**S1 Dataset.**
(XLSX)

**S2 Dataset.**
(XLSX)

## Author Contributions

**Conceptualization:** Samuel Owiti, Denis Hauw.

**Data curation:** Samuel Owiti.

**Formal analysis:** Samuel Owiti.

**Investigation:** Samuel Owiti.

**Methodology:** Samuel Owiti, Denis Hauw.

**Project administration:** Samuel Owiti, Denis Hauw.

**Resources:** Denis Hauw.

**Validation:** Samuel Owiti.

**Writing – original draft:** Samuel Owiti.

**Writing – review & editing:** Samuel Owiti, Denis Hauw.

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
