## [Decision Letter · Decision Letter 0]

10 Apr 2023

PONE-D-23-03165The Initial Development and Validation of the Social Adaptability Skills Questionnaire-SASQPLOS ONE Dear Dr. Owiti,

Thank you for submitting your manuscript to PLOS ONE. After careful consideration, we feel that it has merit but does not fully meet PLOS ONE’s publication criteria as it currently stands. Therefore, we invite you to submit a revised version of the manuscript that addresses the points raised during the review process.

Please see the comments from one reviewer below. The reviewer has commented particularly on the statistics and data analysis, and would like to see these addressed before further review.

Please note that we have only been able to secure a single reviewer to assess your manuscript. We are issuing a decision on your manuscript at this point to prevent further delays in the evaluation of your manuscript. Please be aware that the editor who handles your revised manuscript might find it necessary to invite additional reviewers to assess this work once the revised manuscript is submitted. However, we will aim to proceed on the basis of this single review if possible. 

We look forward to receiving your revised manuscript.

Kind regards,

Hanna Landenmark

Staff Editor

PLOS ONE

Journal Requirements:

a) Did participants provide their written or verbal informed consent to participate in this study?

Reviewers' comments:

Reviewer's Responses to Questions

**Comments to the Author**

1. Is the manuscript technically sound, and do the data support the conclusions?

Reviewer #1: No

2. Has the statistical analysis been performed appropriately and rigorously? 

Reviewer #1: No

3. Have the authors made all data underlying the findings in their manuscript fully available?

Reviewer #1: No

4. Is the manuscript presented in an intelligible fashion and written in standard English?

Reviewer #1: Yes

5. Review Comments to the Author

Reviewer #1: - line 58: please cite more than one study since you wrote "studies".

- line 61: such as? clarify sentence and add citation.

- line 78: indicate citation.

- line 85: idem.

- line 222: why did the authors changed the likert-type scale? please clarify.

- line 249: did the auhtors conducted a priori sample calculations? of not, why not?

- how was the data collected? online questionnaire or paper-and-pencil style? please clarify for replicability.

- table 2: report Skewness and Kurtosis and missing values for each item for transperancy.

- at this point, we are unable to know which items refer to which factor. this needs to be described in full.

- how many factors did the EFA extracted? graph should be disclosed.

- table 3. there are several cross-loadings. how did the authors handled such issues?

- line 331: alfa for which factor? general? this needs to be clarified.

- line 347-350: this section lacks significant information regarding CFA. First, the fit indexes and cutoffs should be reported. Estimator selected should be disclosed. Again, sample size calculations should be reported.

- lines 360-365: here are several issues. In regard to model fit indices, it also seems that the cut-off scores that are being used for your models is too loose. Please check the recommendations by Hu and Bentler (1999) regarding the subject. Also, please update your fit criteria to also include more modern approaches. See Peugh and Feldon (2020).

- Figure 1. also several problems. first, correlating measurement errors is not a acceptable practice as it forces the model to fit. Second, Covariances are above 1, meaning that a) the authors reporte unstandardized loadings b) the model does not fit and variances are extremely high. Third, several low factor loadings are reported (<0.50) meaning that there is insuficient explained variance by these items on the intended factor. Fourth, two items are insuficient to lprovide enough reliability and validity of the factor (see Hair et al., 2019).

- i will not comment further as this study needs substantial revisions.

6. PLOS authors have the option to publish the peer review history of their article (what does this mean?). If published, this will include your full peer review and any attached files.

Reviewer #1: No

---

## [Author Response · Author response to Decision Letter 0]

24 Apr 2023

To the Editor: Thanks for your insightful comments about the current manuscript. We have adjusted the manuscript as per your requests.

To the Reviewer 1: Thanks for taking your time to read and give positive comments which we believe will improve the manuscript.

---

## [Decision Letter · Decision Letter 1]

6 Jun 2023

PONE-D-23-03165R1The Initial Development and Validation of the Social Adaptability Skills Questionnaire-SASQPLOS ONE

Dear Dr. Owiti,

Thank you for submitting your manuscript to PLOS ONE. After careful consideration, we feel that it has merit but does not fully meet PLOS ONE’s publication criteria as it currently stands. Therefore, we invite you to submit a revised version of the manuscript that addresses the points raised during the review process.

**Please note that in light of Reviewer 2's criticism, instead of rejecting your paper, I consider this a risky major revision without a guarantee of publication. Therefore, do your best to convince Reviewer 2, who is a highly regarded expert in psychometrics, scale development and multivariate research, that your revision is at a publishable level, as this might be the last chance for you to do so.**

We look forward to receiving your revised manuscript.

Kind regards,

Ali B. Mahmoud, Ph.D.

Academic Editor

PLOS ONE

Reviewers' comments:

Reviewer's Responses to Questions

**Comments to the Author**

1. If the authors have adequately addressed your comments raised in a previous round of review and you feel that this manuscript is now acceptable for publication, you may indicate that here to bypass the “Comments to the Author” section, enter your conflict of interest statement in the “Confidential to Editor” section, and submit your "Accept" recommendation.

Reviewer #1: All comments have been addressed

Reviewer #2: (No Response)

2. Is the manuscript technically sound, and do the data support the conclusions?

Reviewer #1: Yes

Reviewer #2: No

3. Has the statistical analysis been performed appropriately and rigorously? 

Reviewer #1: Yes

Reviewer #2: No

4. Have the authors made all data underlying the findings in their manuscript fully available?

Reviewer #1: No

Reviewer #2: (No Response)

5. Is the manuscript presented in an intelligible fashion and written in standard English?

Reviewer #1: Yes

Reviewer #2: (No Response)

6. Review Comments to the Author

Reviewer #1: The authors did a good job in reviewing their manuscript. I have no further comments. You revised your manuscript according to my comments.

Reviewer #2: MAJOR ITEMS

-Factor analysis in covariance-based SEM should not be applied to non-reflective sets of measures. Conducting a factor analysis is required for reflective measurement, because reflective constructs assume strong inter-item correlation, and factor analysis is conducted using shared variance (i.e., covariance) between variables (Jarvis et al., 2003; Petter et al., 2007). However, this is not the case with formative measurement, where items within formative constructs are not required to correlate; thus, factor analysis should not include formatives (Cenfetelli & Bassellier, 2009; Petter et al., 2007). If formative measures are included in a factor analysis, the measures will often not factor together reliably because they do not need to correlate. Scholars often resort to eliminating items until they have a correlated set that factor reliably – as has been done in this study (also likely the main cause of the unbalanced final factors, ranging from 2 to 9 items each). This undermines factor content validity for formatively measured constructs because it removes unique dimensions of the construct (Jarvis et al., 2003). Therefore, having sets of formative measures in a covariance-based factor analysis is statistically illogical. To illustrate this with your measures, let’s choose the first factor (“coach”) as an example. First, covariance-based approaches, like factor analysis, group items together based on their shared variance along a scale. Thus, an increase in the scale value should be consistent with an increase in the trait value. However, you cannot have more or less “coach” the way you have labeled and measured it. What trait is increasing or decreasing with this set of measures? Can you put a name on it? If there is not a coherent dimension being represented here, then the factor lacks face validity and lacks any sort of interpretability or meaning. What does it mean when the center of this latent factor increases or decreases? What do you have more or less of? Not coach. Not support from coach. Not positive attitude toward coach. Not adaptability with a new coach. It is too many conceptual dimensions in a single factor (statistical dimension). Beyond this, many of these items are conceptually positive (e.g., “I always feel stimulated whenever my trainer/coach puts pressure on me to reach my sporting goals.”) while many are conceptually negative (“I always feel paralysed whenever my trainer/coach puts pressure on me to reach my sporting goals.”). So, the fact that they move together is curious, but further complicates interpretation of movement on the scale. Metrics of convergent validity and reliability are also misaligned with non-reflective sets of measures. So, the Cronbach’s alpha and AVE (not reported…) are not relevant.

-So, all of that above is to say that SEM and factor analysis are not suitable tools for validating this set of measures. Instead, the authors can perhaps create some sort of inventory or index to indicate the extent to which the athlete is able to adapt to changes in coaches, teammates, clubs, and family proximity. Such indices or inventories are used all the time in psychology. The point of these inventories is to identify areas of strength and weakness in order to tailor an individual treatment plan for the athlete. Sometimes these inventories have reflective dimensions (like the big 5 inventory), and sometimes they don’t (like the Meyers-Briggs Type Indicator). Your set of measures is more like MBTI than Big5. I think the sports motivation scale is similarly measured – i.e., with distinct components that reveal some overall areas for focus, rather than unidimensional reflective factors.

-The sample size for Study 1 was not justified. Only 20 interviews is a pretty small sample. Was this number determined by convenience, a priori, by content saturation (probably the best method for new scale development), or by some other means? Additionally, were these specific 20 individuals selected to maximize variance in experiences – in order to maximize the potential to capture a generalizable set of measures? If not, why then these 20? What might have been missed due to homogeneity of experience? The entire agenda is based on this foundation.

-Why was no attention given to the exorbitant skewness on Items 11-12?

MINOR ITEMS

-I think we might be missing some information here for Study 1: “Items that were rated 4: relevant or less were discussed by the whole panel (4). At the end of this stage, these items were marked for deletion.” This makes it sound like anything that didn’t receive a 5 was deleted. However, what then was the point of discussion? Were ratings adjusted during discussion? Were wording changes made and then items reevaluated? Please clarify.

-Shortly after this, the authors state: “five additional items were added”. How were these items developed? Why were they added? Were they derived from the interviews? Or were they to fill wholes not captured by the interviews? If so, does this point again to the concern I have about representation in the small sample?

7. PLOS authors have the option to publish the peer review history of their article (what does this mean?). If published, this will include your full peer review and any attached files.

Reviewer #1: No

Reviewer #2: No

---

## [Author Response · Author response to Decision Letter 1]

12 Jul 2023

Dear Reviewer 2

Thanks in advance for your insightful comments. We have adjusted the manuscript taking into consideration your valued comments.

All the responses to your questions can be found on the rebuttal letter (file).

Kind regards

---

## [Decision Letter · Decision Letter 2]

25 Jul 2023

PONE-D-23-03165R2The Initial Development and Validation of the Social Adaptability Skills Questionnaire-SASQPLOS ONE

Dear Dr. Owiti,

Thank you for submitting your manuscript to PLOS ONE. After careful consideration, we feel that it has merit but does not fully meet PLOS ONE’s publication criteria as it currently stands. Therefore, we invite you to submit a revised version of the manuscript that addresses the points raised during the review process.

We look forward to receiving your revised manuscript.

Kind regards,

Ali B. Mahmoud, Ph.D.

Academic Editor

PLOS ONE

Reviewers' comments:

Reviewer's Responses to Questions

**Comments to the Author**

1. If the authors have adequately addressed your comments raised in a previous round of review and you feel that this manuscript is now acceptable for publication, you may indicate that here to bypass the “Comments to the Author” section, enter your conflict of interest statement in the “Confidential to Editor” section, and submit your "Accept" recommendation.

Reviewer #1: All comments have been addressed

Reviewer #2: (No Response)

2. Is the manuscript technically sound, and do the data support the conclusions?

Reviewer #1: Yes

Reviewer #2: Partly

3. Has the statistical analysis been performed appropriately and rigorously? 

Reviewer #1: Yes

Reviewer #2: No

4. Have the authors made all data underlying the findings in their manuscript fully available?

Reviewer #1: Yes

Reviewer #2: Yes

5. Is the manuscript presented in an intelligible fashion and written in standard English?

Reviewer #1: Yes

Reviewer #2: Yes

6. Review Comments to the Author

Reviewer #1: the authors revised the manuscript accordingly. The revisions are worthy of attention and they revised the manuscript according to the reviewer comments.

Reviewer #2: -Thank you for your thoughtful consideration of my feedback and your extensive edits to address my concerns. I appreciate the improvements.

-One of the remaining concerns I have is either large or small, but without clarification, it is hard to know. The authors stated that the instrument is a mix of positively and negatively worded items. However, they did not state whether they reversed the values of the negatively worded items in order to have them positively correlate with the other items. From their rotated factor matrix, it appears they probably did reverse them since all loadings are positive. If they weren’t reversed, then items 8 and 13 should be inversely correlated (among others). If the authors did reverse the negatively worded items’ values, this should be clearly stated. If they did not reverse them, then we have a bigger issue, since this implies the data is inconsistent (with participants responding the same way to both positive and negative wording).

-The other concern is with regards to the use of CFA to validate these groups. As was clearly shown in the CFA, the loadings were weak and certainly would not pass criteria of convergent validity. This completely undermines the primary objective of this study.

-Issues of discriminant validity were also not mentioned; however this is critical to an assessment via factor analysis.

7. PLOS authors have the option to publish the peer review history of their article (what does this mean?). If published, this will include your full peer review and any attached files.

Reviewer #1: No

Reviewer #2: No

---

## [Author Response · Author response to Decision Letter 2]

31 Jul 2023

1. Did the authors reverse the negatively worded items’ values, this should be clearly stated:

Yes, we did reverse the negatively worded questions in SPSS before analysis (i.e., 1 = 6, 2 = 5, 3 = 4, 5 = 2, and lastly 6 = 1). The following phrase has been added on the manuscript:

-Prior to the analysis, scores of negatively worded questions were reversed.

2. Convergent analysis missing

We have carried a CV analysis and additional paragraphs added on the manuscript.

Since we had to use the criterion values in reporting the CV. The literature reports no consensus as regards the best criterion. Several studies have reported different values with some reporting either “low” or “high” CV while criterion between r = .28 and r = .75 being accepted.

Results

The measures of family and club constructs (0.761 and 0.732) respectively reported high convergent validity. In contrast, both the measures of coach and teammates constructs (0.631 and 0.657) respectively reported average or modest convergent validity.

The following paragraph has been added to the manuscript to argue for the debatable CV criterion levels:

Convergent validities above r = .70 are recommended, whereas those below r = .50 should be avoided (Carlson et al., 2012). However, research evidence suggests that actual levels of convergent validity in psychology research still vary widely (Carlson et al., 2012). Therefore, without more specific guidance, researchers reach logically inconsistent conclusions, arguing that a convergent validity as low as r = .28 (e.g., Larraza-Kintana, Wiseman-Meija & Welbourne, 2007) indicates the measures converge, whereas others report convergent analysis as high as r = .75 (e.g., Podsakoff, Whiting, Podsakoff & Baume, 2018) signalling high convergence.

3. Discriminant validity analysis not mentioned

-Thanks for the comment; 

We have now performed a discriminant validity analysis using 

the heterotrait-monotrait ratio of correlations (HTMT) as a new approach to assess discriminant validity in variance-based SEM (Henseler, Ringle & Sarstedt, 2015). This is due to the fact that recent research suggests that the routinely used Fornell-Larcker criterion is not effective under certain circumstances of discriminant validity testing (Henseler et al. 2014).

-HTMT procedure can be assessed through two ways (i) as a criterion, and (ii) a statistical test. We chose the latter due to the following reasons: (a) predefined criterion thresholds are debatable (Clark & Watson, 1995; Kline, 2016), and (b) statistical test involves bootstrapping which allows for constructing confidence intervals and also reducing familywise error rate.

Results

The computations yields values between (0.661 and 0.849) (Table 4). Upon comparing if the CI result value (one) falls outside the interval range, it can be reported that all the four constructs are empirically distinct.

---

## [Decision Letter · Decision Letter 3]

2 Aug 2023

The Initial Development and Validation of the Social Adaptability Skills Questionnaire-SASQ

PONE-D-23-03165R3

Dear Dr. Owiti,

We’re pleased to inform you that your manuscript has been judged scientifically suitable for publication and will be formally accepted for publication once it meets all outstanding technical requirements.

Kind regards,

Ali B. Mahmoud, Ph.D.

Academic Editor

PLOS ONE

Additional Editor Comments (optional):

Reviewers' comments:

Reviewer's Responses to Questions

**Comments to the Author**

1. If the authors have adequately addressed your comments raised in a previous round of review and you feel that this manuscript is now acceptable for publication, you may indicate that here to bypass the “Comments to the Author” section, enter your conflict of interest statement in the “Confidential to Editor” section, and submit your "Accept" recommendation.

Reviewer #2: All comments have been addressed

2. Is the manuscript technically sound, and do the data support the conclusions?

Reviewer #2: (No Response)

3. Has the statistical analysis been performed appropriately and rigorously? 

Reviewer #2: (No Response)

4. Have the authors made all data underlying the findings in their manuscript fully available?

Reviewer #2: (No Response)

5. Is the manuscript presented in an intelligible fashion and written in standard English?

Reviewer #2: (No Response)

6. Review Comments to the Author

Reviewer #2: (No Response)

7. PLOS authors have the option to publish the peer review history of their article (what does this mean?). If published, this will include your full peer review and any attached files.

Reviewer #2: No

---

## [Editor Report · Acceptance letter]

10 Aug 2023

PONE-D-23-03165R3 

The Initial Development and Validation of the Social Adaptability Skills Questionnaire: SASQ 

Dear Dr. Owiti:

I'm pleased to inform you that your manuscript has been deemed suitable for publication in PLOS ONE. Congratulations! Your manuscript is now with our production department. 

Kind regards, 

on behalf of

Dr. Ali B. Mahmoud 

Academic Editor

PLOS ONE